# Predictors of Opioid Prescribing for Non-Malignant Low Back Pain in an Italian Primary Care Setting

**DOI:** 10.3390/jcm10163699

**Published:** 2021-08-20

**Authors:** Simona Cammarota, Valeria Conti, Graziamaria Corbi, Luigi Di Gregorio, Pasquale Dolce, Marianna Fogliasecca, Teresa Iannaccone, Valentina Manzo, Vincenzo Passaro, Bernardo Toraldo, Alfredo Valente, Anna Citarella

**Affiliations:** 1LinkHealth Health Economics, Outcomes & Epidemiology s.r.l., 80143 Naples, Italy; marianna.fogliasecca@linkhealth.it (M.F.); anna.citarella@linkhealth.it (A.C.); 2Clinical Pharmacology and Pharmacogenetics Unit, University Hospital “San Giovanni di Dio e Ruggi d’Aragona”, 84084 Salerno, Italy; vconti@unisa.it (V.C.); tiannaccone@unisa.it (T.I.); vmanzo@unisa.it (V.M.); 3Department of Medicine, Surgery and Dentistry “Scuola Medica Salernitana”, University of Salerno, 84081 Baronissi, Italy; a.valente@studenti.unisa.it; 4Department of Medicine and Health Sciences, “Vincenzo Tiberio”, University of Molise, 86100 Campobasso, Italy; graziamaria.corbi@unimol.it; 5Italian Society of Gerontology and Geriatrics (SIGG), 50122 Florence, Italy; 6Parmenide Medical Cooperative, 84084 Salerno, Italy; gigidoc50@gmail.com (L.D.G.); studiovpassaro@gmail.com (V.P.); 7Department of Public Health, University of Naples “Federico II”, 80138 Naples, Italy; pasquale.dolce@unina.it; 8Alfasigma Italia, 40133 Bologna, Italy; bernardo.toraldo@alfasigma.com

**Keywords:** opioid, non-steroidal anti-inflammatory drugs, lower back pain, primary care

## Abstract

This study explores which patient characteristics could affect the likelihood of starting low back pain (LBP) treatment with opioid analgesics vs. Non-Steroidal Anti-Inflammatory Drugs (NSAIDs) in an Italian primary care setting. Through the computerized medical records of 65 General Practitioners, non-malignant LBP subjects who received the first pain intensity measurement and an NSAID or opioid prescription, during 2015–2016, were identified. Patients with an opioid prescription 1-year before the first pain intensity measurement were excluded. A multivariable logistic regression model was used to determine predictive factors of opioid prescribing. Results were reported as Odds Ratios (ORs) with a 95% confidence interval (CI), with *p* < 0.05 indicating statistical significance. A total of 505 individuals with LBP were included: of those, 72.7% received an NSAID prescription and 27.3% an opioid one (64% of subjects started with strong opioid). Compared to patients receiving an NSAID, those with opioid prescriptions were younger, reported the highest pain intensity (moderate pain OR = 2.42; 95% CI 1.48–3.96 and severe pain OR = 2.01; 95% CI 1.04–3.88) and were more likely to have asthma (OR 3.95; 95% CI 1.99–7.84). Despite clinical guidelines, a large proportion of LBP patients started with strong opioid therapy. Asthma, younger age and pain intensity were predictors of opioid prescribing when compared to NSAIDs for LBP treatment.

## 1. Introduction

Over the past decade, opioid prescribing for Chronic Non-Cancer Pain (CNCP) has increased in many countries despite limited supporting evidence, high costs, and often related serious health risks [1,2]. Low Back Pain (LBP) is one of the most common CNCP conditions for which opioids are prescribed in the primary care setting [3,4,5]. This opioid prescribing frequency is particularly worrying as LBP is commonly reported by individuals visiting their General Practitioners (GPs). Globally, LBP is the leading cause of disability, activity limitation, and absenteeism from work, with a high medical load and economic costs [6]. Since the relevant LBP burden, several guidelines have been issued for the treatment, specifically discouraging opioid use. The “CDC Guideline for Prescribing Opioids for Chronic Pain—United States, 2016” underlines the harms associated with opioid use and recommends pursuing alternative options for managing CNCP [1]. The 2017 American College of Physicians guidelines recommend that physicians should consider treatment with Non-Steroidal Anti-Inflammatory Drugs (NSAIDs) as first-line therapy, and tramadol or duloxetine as second-line therapy, in patients with chronic LBP who have had an inadequate response to non-pharmacologic therapy [7]. The 2016 NICE guidelines suggest weak opioids as a second-line treatment when NSAIDs are contraindicated, ineffective, or not tolerated [8]. All the aforementioned guidelines indicate that opioids should not be considered as an option for LBP unless recommended treatments fail and only if the potential benefits are expected to outweigh the risks. In Italy, together with the assessment of patient characteristics, pain severity constitutes the first criterion to guide the selection of the most appropriate analgesic drug [9]. Despite this, a recent study in an Italian primary care setting revealed that strong opioids were the second most-prescribed drugs following NSAIDs, also in the cohort of LBP patients with mild pain severity [10]. However, the factors that lead GPs to initiate opioid treatment in patients with LBP are still not fully understood. Understanding these factors with regards to pain management for LBP can help focus quality improvement efforts and improve the development of quality measures.

Therefore, this study aims to explore which patient characteristics could affect the likelihood of starting LBP treatment with opioid analgesics vs. NSAIDs in an Italian primary care setting.

## 2. Methods

### 2.1. Data Source and Study Population

An observational retrospective study was carried out with the cooperation of 65 GPs, (covering an assisted population of 90,000, Campania region), all belonging to the Italian National Health Service (NHS) and affiliated to the “Parmenide” medical association. All the participating GPs used the same software to record data during their daily practice and received formal periodic training for data entry. For the purpose of this study, data were retrieved using an anonymous encrypted patient code linking demographic details with medical diagnoses, drug prescriptions and date of death or censoring (transfer to another GP). All diagnoses were coded according to the International Classification of Diseases, 9th Revision, Clinical Modification (ICD-9-CM) and the Anatomical Therapeutic Chemical (ATC) classification, respectively.

Individuals aged 18 years and over with a diagnosis of LBP or sciatica (ICD9 codes 724.1, 724.2, 724.3, 724.4, 724.5), who received the pain intensity measurement and an NSAID (ATC M01A) or opioid prescription (weak opioid ATC N02AA59, N02AX52, N02AX02 or strong opioid ATC N02AA01, N02AA05, N02AX06, N02AB03), between February 1, 2015 and January 31, 2016, were included. The date of the first pain intensity measurement during the study period was defined as the index date. The 365 days prior to the index date were checked to exclude patients who had received an opioid prescription during that period. Moreover, since we aimed to investigate the treatment in LBP patients without a cancer diagnosis, individuals with a diagnosis of primary or metastatic cancer (140.x–209.xx, 230.x–239.x) at any time before the index date were excluded.

Finally, subjects were divided into two mutually exclusive groups: (i) those receiving an NSAID prescription at the index date (NSAID group); (ii) those receiving an opioid prescription with or without NSAIDs (Opioid group).

### 2.2. Patient Characteristics

For each patient, the following variables were assessed at the index date: age groups (≤45, 45–54, 55–64, 65–74, ≥75), sex, pain intensity measurement and comorbidities. The pain intensity was measured by the GPs using the 11-point Numerical Rating Scale (NRS-11). Specifically, each patient was asked to express the intensity of their pain with a number ranging from 0 (no pain) to 10 (the worst possible pain). The following comorbidities assessed at the index date were included in the analysis: diabetes, hypertension, cardiovascular or cerebrovascular diseases, chronic obstructive pulmonary disease (COPD), asthma, gastrointestinal disorders, chronic renal disease, arthropathy, rheumatism, mental disorders, and depression.

### 2.3. Statistical Analysis

Categorical variables were summarized using frequencies and percentages; continuous variables were summarized using means and standard deviations since the distributions were symmetrical. For the univariate analysis, we used χ^2^ tests to compare categorical variables between the groups and the Student’s t test for independent samples to compare continuous variables between the groups. A multivariable logistic regression model was used to determine the independent predictive factors of opioid prescribing. The dependent variable was opioid vs. NSAID prescribing. Age group, sex, pain intensity (mild pain (NRS 1–4), moderate pain (NRS 5–7), severe pain (NRS 8–10)) and comorbidities were the independent factors assessed. Multicollinearity was assessed computing the Variance Inflation Factor (VIF). The results were reported as Odds Ratio (OR) with 95% confidence interval (CI) and a *p*-value < 0.05 was considered statistically significant. All analyses were conducted using the SPSS software (SPSS Inc. version 23.0, Chicago, IL, USA).

## 3. Results

A final sample of 505 patients who met the inclusion criteria was identified: 367 (72.7%) in the NSAID group and 138 (27.3%) in the opioid group. Table 1 reports patients’ characteristics of both study groups. Most patients were female, 58.9% in the NSAID group and 56.5% in the opioid group. Compared to patients receiving NSAID prescriptions, those with opioids were younger (mean age of 56.4 vs. 60.8; *p* = 0.005) and reported higher NRS-11 (5.6 vs. 5.1; *p* = 0.007). Regarding comorbidities, statistically significant differences between the two groups were found only for gastrointestinal disorders (4.3% opioid group vs. 16.1% NSAID group, *p* < 0.0001) and asthma (18.1% vs. 6.5%, *p* < 0.0001).

The 24.6% in the opioid group and 63.2% in the NSAID group (*p* < 0.0001) received at least 1 prescription for an NSAID 1-year before the index date. In both groups, diclofenac was the most prescribed NSAID followed by ketoprofen, celecoxib, and ibuprofen (Figure 1).

Regarding the type of opioid analgesic, 63.8% of patients received a prescription for a strong opioid and 36.2% for a weak opioid. Of these, an oxycodone/naloxone combination was the most frequently prescribed (57.3%), followed by codeine combinations (29.0%). Only a minority of patients received a tramadol (7.2%), tapentadol (5.8%), and fentanyl (0.7%) prescription (Figure 2).

The demographic and clinical factors associated with the likelihood of receiving an opioid prescription are shown in Table 2. No collinearity issue was found, since all variables had a value of VIF well below 5 (maximum value was 1.89 for age). Patients were more likely to receive an opioid if they referred higher NRS during the visit (moderate pain OR = 2.42; 95%CI 1.48–3.96 and severe pain OR = 2.01;95% CI 1.04–3.88), whereas age was negatively associated with opioid prescriptions (Table 2) Moreover, the odds of belonging to the opioid group were 3.95 (95% CI 1.99–7.84) for patients with asthma, while patients with gastrointestinal disorders had lower odds of receiving an opioid prescription (OR = 0.22; 95% CI 0.09–0.55).

## 4. Discussion

Through an analysis of demographic and clinical conditions, as well as pain severity, our study explores the factors which lead GPs to initiate opioid treatment in patients with non-cancer LBP. Our findings show few factors associated with the higher likelihood of receiving opioid prescriptions when compared to NSAIDs drugs. As expected, we found that the increasing pain intensity was positively associated with opioid prescriptions. Patients with moderate or severe pain had a 2-fold increase in risk of receiving opioids compared to those with mild pain. However, this does not mean that pain intensity is the only factor that physicians should consider. In prescribing opioids for LBP, GPs appear to be influenced by factors unrelated to pain intensity. Interestingly, we found that the diagnosis of asthma was strongly associated with opioid prescriptions. Specifically, when compared with NSAIDs, the odds of receiving an opioid prescription were almost four-fold higher for patients with asthma after adjustment for all other variables. Our results suggest that GPs should have a higher index of suspicion for NSAID-exacerbated respiratory disease in subjects with asthma. A well-known distinct clinical syndrome could be induced by the increased leukotriene synthesis in response to Cyclo-Oxygenases (COX) inhibition by NSAIDs. Particularly, COX-1 may mediate bronchoconstriction and inflammatory changes in asthma pathophysiology [11,12]. This could lead physicians to avoid the use of NSAIDs in patients with asthma. However, as reported in several studies, highly specific COX-2 inhibitors should be well-tolerated and could be safely used in asthmatic patients [13].

Based on the odds ratio by the multivariable analysis, age also appears to play a significant role. Older patients were significantly less likely to receive opioids and were more likely to receive NSAIDs. This could be related to the well-known increased responsiveness to the effects of opioids in the elderly and the polypharmacy-related potential risk of drug–drug interactions [14]. This may result in a raised risk of side effects, including sedation and mild cognitive impairment, severe constipation, urinary retention, and respiratory depression [15]. Increased sensitivity of older patients to systemic opioids mostly involves pharmacokinetic factors such as a higher proportion of unbound and active substances, as well as changes in drug redistribution. Because of a 40% age-related reduction in stroke volume in the elderly, there is a protracted redistribution of opioids to the liver, hesitating in a prolonged time of liver metabolization, with lesser inactivation over time, and a consequent increased duration of action [16]. Furthermore, a recent systemic review with metanalysis confirms that opioid use is also associated with falls, fall-related injuries, and fractures among the elderly population [17].

In our study, approximately 64% of patients in the opioid group started with a strong opioid. Although these patients might simply have failed to be managed with alternative therapies or did not tolerate drugs like NSAIDS, the current guidelines recommend strong opioid prescriptions only for unremitting cases and even for short-term use, stepping patients down to weaker opioids where appropriate or removal altogether if not effective [8,16].

Our results should be interpreted in the context of some limitations. First, we could not differentiate between acute, subacute, and chronic LBP as well as nociceptive, neuropathic, and mixed pain because data were not available in the database under analysis. Second, subjects treated with OTC (Over The Counter) oral analgesics, such as paracetamol and NSAIDs, may have been underestimated, since these drugs can be directly purchased by the patient; therefore, they were not tracked in the database under analysis. Finally, generalizations of the study’s findings should be made with caution, particularly because the participating GPs were all from a Southern Italian area, and therefore were not fully representative of the GP workforce across Italy.

In conclusion, our study shows that asthma, younger age and pain intensity are the factors leading GPs to initiate opioid treatment in patients with non-cancer LBP. Moreover, despite clinical guidelines suggesting that opioids should not be considered an option for LBP unless recommended treatments fail and only if the potential benefits are expected to outweigh the risks, a large proportion of patients are starting treatment with a strong opioid.

## Figures and Tables

**Figure 1 jcm-10-03699-f001:**
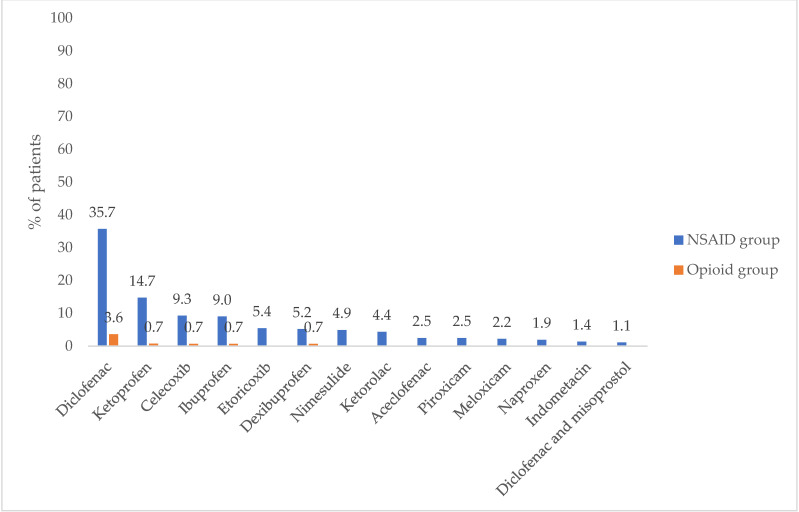
Percentage of patients with NSAID prescriptions. Abbreviations: NSAIDs, Non-Steroidal Anti-Inflammatory Drugs.

**Figure 2 jcm-10-03699-f002:**
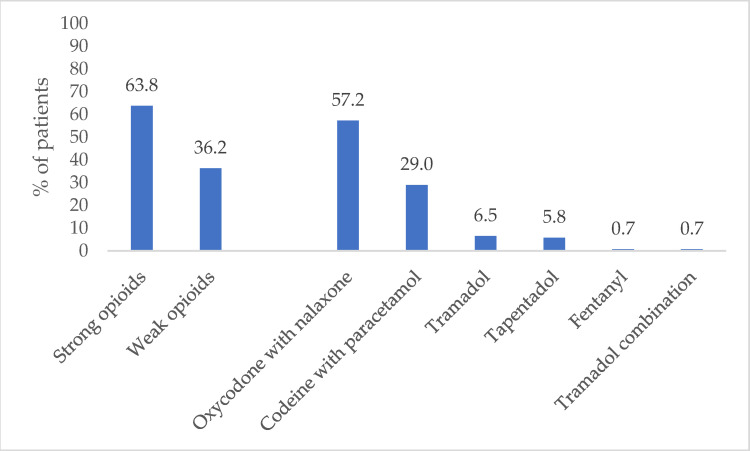
Percentage of patients with an opioid prescription.

**Table 1 jcm-10-03699-t001:** Characteristics of patients with LBP receiving Non-Steroidal Anti-Inflammatory Drugs (NSAID group) or opioid prescriptions (opioid group).

	NSAID Group (N = 367)	Opioid Group (N = 138)	*p* Value
**Female gender, *n* (%)**	216 (58.9)	78 (56.5)	0.64
**Age in years, mean ± SD**	60.8 ± 15.2	56.4 ± 17.2	**0.005**
**Age groups, *n* (%)**			
<45	49 (13.4)	38 (27.5)	**0.002**
45–54	73 (19.9)	30 (21.7)	
55–64	96 (26.2)	23 (16.7)	
65–74	68 (18.2)	23 (16.7)	
≥75	81 (22.1)	24 (17.4)	
**Pain intenstity**			
NRS, mean ± SD	5.1 ± 2.0	5.6 ± 2.0	**0.007**
Mild pain (NRS 1–4), *n* (%)	166 (45.2)	37 (26.8)	0.001
Moderate pain (NRS 5–7), *n* (%)	152 (41.4)	77 (55.8)	
Severe pain (NRS 8–10), *n* (%)	49 (13.4)	24 (17.4)	
**Comorbidities, *n* (%)**			
Hypertension	198 (54.0)	63 (45.7)	0.10
Arthropathies	197 (53.7)	61 (44.2)	0.06
Rheumatic diseases	52 (14.2)	25 (18.1)	0.27
Diabetes	64 (17.4)	23 (16.7)	0.84
Neck pain	58 (15.8)	32 (23.2)	0.05
Gastrointestinal Disorders	59 (16.1)	6 (4.3)	**<0.0001**
COPD	45 (12.3)	17 (12.3)	0.99
Cardio- or Cerebrovascular diseases	35 (9.5)	18 (13.0)	0.25
Asthma	24 (6.5)	25 (18.1)	**<0.0001**
Mental disorders	25 (6.8)	10 (7.2)	0.86
Chronic renal failure	5 (1.4)	4 (2.9)	0.24
**NSAIDs before index date, *n* (%)**	232 (63.2)	34 (24.6)	**<0.0001**
**NSAIDs before index date, mean ± SD**	2.30 ± 3.44	0.85 ± 2.49	**<0.0001**

Abbreviations: LBP, low back pain; COPD, chronic obstructive pulmonary disease; NRS, Numeric Rating Scale; NSAIDs, Non-Steroidal Anti-Inflammatory Drugs; SD, Standard Deviation. Bold values denote statistical significance (*p* < 0.05).

**Table 2 jcm-10-03699-t002:** Clinical and demographic factors associated with an opioid prescription (ref. NSAID).

Covariates	Adjusted Odds Ratio	95% CI	*p* Value
**Female gender (ref. male)**	1.09	0.69–1.74	0.71
**Age groups, (ref. < 45 years)**			
45–54	**0.29**	**0.12–0.71**	**0.007**
55–64	**0.50**	**0.25–0.97**	**0.42**
65–74	**0.26**	**0.12–0.56**	**0.001**
≥75	**0.38**	**0.16–0.88**	**0.25**
**Pain intensity (ref. Mild pain (NRS 1–4))**			
Moderate pain (NRS 5–7)	**2.42**	**1.48–3.96**	**0.0001**
Severe pain (NRS 8–10)	**2.01**	**1.04–3.88**	**0.38**
**Comorbidity (ref. no comorbidity)**			
Hypertension	1.15	0.65–2.05	0.11
Arthropathies	0.98	0.58–1.61	0.94
Rheumatic diseases	1.54	0.82–2.90	0.18
Diabetes	1.15	0.61–2.14	0.67
Neck pain	1.56	0.90–2.70	0.07
Cardio-or Cerebro-vascular diseases	1.97	0.93–4.16	0.07
Asthma	**3.95**	**1.99–7.84**	**0.0001**
COPD	0.81	0.40–1.64	0.56
Chronic renal failure	1.37	0.29–6.50	0.69
Gastrointestinal Disorders	**0.22**	**0.09–0.55**	**0.001**
Mental disorder/depression	1.34	0.53–3.37	0.54

Abbreviations: COPD, chronic obstructive pulmonary disease; NSAIDs, Non-Steroidal Anti-Inflammatory Drugs; CI, Confidence Interval. Bold values denote statistical significance (*p* < 0.05).

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
