# Peer review of "Predictors of Opioid Prescribing for Non-Malignant Low Back Pain in an Italian Primary Care Setting"

_jcm, 2021, doi:10.3390/jcm10163699_

Round 1

Reviewer 1 Report

this study has a great conclusion important for everyone

how this is reached could be moore sufficient; by dividing NRS scores betwwn 1-5/5-7/8-10 and age in decades the evidence will be much higher and the differences between groups bigger and morde meaningfull

Author Response

Response to Reviewer 1 Comments

Point 1: this study has a great conclusion important for everyone how this is reached could be more sufficient; by dividing NRS scores between 1-5/5-7/8-10 and age in decades the evidence will be much higher and the differences between groups bigger and more meaningful.

As suggested we divided the NRS score in 3 groups 1-4, 5-7, 8-10 and the age in 5 groups (≤45, 45-54, 55-64, 65-74, ≥75). We didn’t use the decades for the youngest and the oldest age groups due to the low frequency. As suggested, the differences between groups are bigger and more meaningful. We updated the table 1 and table 2 using the new categorial variables as well as the results and discussion sections.

Reviewer 2 Report

  1. All abbreviations should be clarified, even LBP.
  2. “Moreover, in order to focus on non-cancer LBP treatment individuals with a diagnosis of primary or metastatic cancer (140.x–209.xx, 230.x–239.x) at any time before the index date were excluded.” This sentence is not fully clear to me. Who was excluded?
  3. As I understand, patients were examined only once in your study “index day”?
  4. Did you assess doctors? Maybe treatment was GPs related and some doctors didn’t prescribe opioids?

Author Response

Point 1: All abbreviations should be clarified, even LBP.

As suggested, the abbreviation were clarified in the text.

 Point 2: “Moreover, in order to focus on non-cancer LBP treatment individuals with a diagnosis of primary or metastatic cancer (140.x–209.xx, 230.x–239.x) at any time before the index date were excluded.” This sentence is not fully clear to me. Who was excluded?

We modified the text in the  methods section to clarify the exclusion criteria as following “Since we aimed to investigate the treatment of LBP patients without cancer diagnosis, we excluded patients with primary or metastatic diagnosis cancer at any time before the index date.”

Point 3: As I understand, patients were examined only once in your study “index day”?

Yes, patients were examined the NSAID or opioid prescription only at the index date, which is the date of the first pain intensity measurement during the study period (February 1, 2015 - January 31, 2016).

Point 4: Did you assess doctors? Maybe treatment was GPs related and some doctors didn’t prescribe opioids?

We agree, it will be interested to assess the doctors effect, however, we don’t have this information in our data.

Round 2

Reviewer 1 Report

no suggestions; well written article

but does it bring much news?